# CDC25C Protein Expression Correlates with Tumor Differentiation and Clinical Outcomes in Lung Adenocarcinoma

**DOI:** 10.3390/biomedicines11020362

**Published:** 2023-01-26

**Authors:** Esther Stern, Guy Pines, Li Or Lazar, Gilad W. Vainer, Nitzan Beltran, Omri Dodi, Lika Gamaev, Ofir Hikri Simon, Michal Abraham, Hanna Wald, Amnon Peled, Ori Wald

**Affiliations:** 1Gene Therapy Institute, Hadassah Hebrew University Medical Center, Faculty of Medicine, Hebrew University of Jerusalem, Jerusalem 91120, Israel; 2Department of General Surgery, Kaplan Hebrew University Medical Center, Faculty of Medicine, Hebrew University of Jerusalem, Rehovot 76100, Israel; 3Department of General Thoracic Surgery, Kaplan Hebrew University Medical Center, Faculty of Medicine, Hebrew University of Jerusalem, Rehovot 76100, Israel; 4Department of Pathology, Hadassah Hebrew University Medical Center, Faculty of Medicine, Hebrew University of Jerusalem, Jerusalem 91120, Israel; 5Department of Pathology, Kaplan Hebrew University Medical Center, Faculty of Medicine, Hebrew University of Jerusalem, Rehovot 76100, Israel; 6Department of Cardiothoracic Surgery, Hadassah Hebrew University Medical Center, Faculty of Medicine, Hebrew University of Jerusalem, Jerusalem 91120, Israel

**Keywords:** NSCLC, CDC25C, lung adenocarcinoma, lung cancer

## Abstract

Given that, even after multimodal therapy, early-stage lung cancer (LC) often recurs, novel prognostic markers to help guide therapy are highly desired. The mRNA levels of cell division cycle 25C (CDC25C), a phosphatase that regulates G2/M cell cycle transition in malignant cells, correlate with poor clinical outcomes in lung adenocarcinoma (LUAD). However, whether CDC25C protein detected by immunohistochemistry can serve as a prognostic marker in LUAD is yet unknown. We stained an LC tissue array and a cohort of 61 LUAD tissue sections for CDC25C and searched for correlations between CDC25C staining score and the pathological characteristics of the tumors and the patients’ clinical outcomes. Clinical data were retrieved from our prospectively maintained departmental database. We found that high expression of CDC25C was predominant among poorly differentiated LUAD (*p* < 0.001) and in LUAD > 1cm (*p* < 0.05). Further, high expression of CDC25C was associated with reduced disease-free survival (*p* = 0.03, median follow-up of 39 months) and with a trend for reduced overall survival (*p* = 0.08). Therefore, high expression of CDC25C protein in LUAD is associated with aggressive histological features and with poor outcomes. Larger studies are required to further validate CDC25C as a prognostic marker in LUAD.

## 1. Introduction

Despite advances in lung cancer (LC) diagnostics and therapeutics, LC remains the leading cause of cancer-related death worldwide, with over 135,000 yearly deaths attributed to the disease in the United States alone [1]. Of all LCs, 84% belong to the histological type of non-small cell lung cancer (NSCLC), with the most common subtypes being lung adenocarcinoma (LUAD) and lung squamous cell carcinoma (LUSC), representing 66% and 33% of all NSCLC, respectively [2]. The outcome of NSCLC patients is primarily determined by stage of disease at time of diagnosis. A total of 70% of NSCLC patients are diagnosed with advanced disease stage, a stage when a cure is nearly unachievable, whereas 30% of patients are diagnosed with localized disease, a clinical scenario in which a cure can still be a therapeutic goal [1]. Surgery, which offers complete resection of the tumor, is a fundamental part of treatment for localized NSCLC; however, even among patients that receive surgery, the disease often recurs [3]. To reduce disease recurrence rates and improve long-term outcomes, neo-adjuvant and/or adjuvant treatments are often administered in conjunction with surgery [4]. Unfortunately, the benefit of these therapies is rather limited and they are associated with significant side effects [5,6]. Whether to administer neo-adjuvant and/or adjuvant treatments or not is a critical clinical question and the decision is primarily based on stage of disease at time of diagnosis [4]. Prognostic markers, other than disease stage, which may aid in discriminating between NSCLC patients that will or will not have disease recurrence are highly desired, in particular, since they may help personalize the decision on neo-adjuvant and/or adjuvant treatments [7].

In search for new prognostic markers for NSCLC, cell cycle regulatory proteins emerge as promising targets for two main reasons: first, dysregulation of their expression or function is considered a hallmark of disease aggressiveness; second, drugs that target them show positive effects on disease outcome in advanced phase trials [8,9,10]. Of the cell cycle proteins, the cell division cycle 25 (CDC25) family of dual specificity phosphatases is specifically interesting as potential new prognostic markers for NSCLC. In particular, because the mRNA levels of all three isoforms of CDC25—namely CDC25A, CDC25B and CDC25C—are elevated in this type of tumor [11]. The main action of the CDC25s is mediated by dephosphorylation of their substrates, cyclin-dependent kinases (CDKs), which, in turn, drive the cell cycle forward. CDC25A mainly acts at the G1/S cell cycle transition, whereas CDC25B and CDC25C play important roles in the G2/M cell cycle transition [12,13,14,15]. Data regarding the mRNA expression levels of the CDC25s in NSCLC and regarding their association with clinical outcomes can be extracted from The Cancer Genome Atlas (TCGA) database [16] and from the human protein atlas database [17]. Remarkably, TCGA data indicate that, whereas mRNAs of all three isoforms of CDC25 are overexpressed in NSCLC [11], only high expression of CDC25C is associated with poor survival (*p* < 0.001; data not shown), with this association being much stronger in LUAD rather than in LUSC (*p* < 0.0001 and *p* < 0.05, respectively; data not shown). Thus, of the CDC25 phosphatases, CDC25C emerges as a potentially new prognostic marker in LUAD. 

Further evidence for the pathologic role of CDC25C in LUAD comes from reports showing that downregulation of CDC25C mRNA in LUAD is associated with suppression of tumor cell proliferation [18] and that upregulation of CDC25C mRNA in the tumor is associated with acquisition of resistance to the anti-LC chemotherapeutic drugs cisplatin [19] and ganetespib [20]. More recently, two studies reported that CDC25C mRNA levels in LUAD influence the tumor immune cell infiltration, as well as its response to immunotherapy. In particular, CDC25C mRNA levels were shown to positively correlate with the infiltration of Th2 T cells to the tumors, while they were shown to negatively correlate with the infiltration of mast cells, eosinophils, natural killer cells and dendritic cells to the tumors [21]. In addition, high CDC25C mRNA levels in the tumor were shown to correlate with expression of genes that drive immune suppression. In line with these observations, elevated levels of CDC25C mRNA in LUAD were also linked to shorter progression-free survival in LUAD patients treated with immunotherapy [22]. From a clinical perspective, these reports suggest that CDC25C mRNA levels in LUAD may potentially guide therapeutic decisions. 

It is evident that, in order to further develop CDC25C as a prognostic marker in LUAD, the findings of previous studies must be extended from the mRNA level to the protein level. In the current work, we address this gap by staining an LC tissue array as well as a cohort of 61 tissue sections derived from a series of LUAD patients that we operated on for CDC25C. We comprehensively characterize CDC25C protein expression patterns in the tumors while also measuring the association between CDC25C expression levels and the disease characteristics and patient clinical outcomes.

## 2. Materials and Methods

### 2.1. Lung Cancer Tissue Array

Assessment of CDC25C expression in LC histological subtypes and analysis of its correlation with disease characteristics were conducted using the Biomax LC tissue array: LC807b. This array contains 70 core tissue samples of a spectrum of lung pathologies, including LUAD *n* = 27 (some of which are termed broncho-alveolar carcinomas *n* = 6), LUSC *n* = 31, adenosquamous carcinoma *n* = 1, large cell carcinomas *n* = 5 and small cell carcinomas *n* = 6, and 10 tissue cores of tumor-adjacent normal lung tissues (Biomax US. 1100 Taft St., Rockville, MD 20850, USA). Table 1 shows the characteristics of the 27 LUAD and 31 LUSC core tumor sections included in the LC array.

### 2.2. Patients and Tissue Collection

We utilized the prospectively maintained database of the department of general thoracic surgery at Kaplan Medical Center (KMC) to retrospectively identify all patients who had a curative intent surgery for resection of NSCLC between January 2012 and September 2018. Among these patients, we particularly searched for patients with a confirmed pathological diagnosis of LUAD, who had not received preoperative chemotherapy or radiotherapy. In total, 61 consecutive cases were identified. We termed them the KMC cohort. For this study, the pathologist at KMC re-reviewed the histological tumor sections of each patient and selected a representative slide from each tumor. The pathologist specifically selected slides that contained both tumor tissue and tumor-adjacent lung parenchyma. We prepared tissue sections from the formalin-fixed and paraffin-embedded tumor blocks that corresponded to the pathologist selection and we then stained them for CDC25C. Table 2 shows the characteristics of the patient and tumors included in the KMC cohort.

The Kaplan Hebrew University Medical Center institutional review board (institutional Helsinki committee) approved this study (protocol number KMC-0233-17). The review board permitted acquisition of surplus histological material for analysis and review of patients’ medical records.

### 2.3. Clinical Data Collection and Determination of Tumor Characteristics in the KMC Cohort

The following clinical parameters were collected for each patient: age, sex, smoking status, forced expiratory volume in 1 s, precent from predicted (FEV1%), chronic obstructive pulmonary disease (COPD) status, diabetes mellitus (DM) status and hemoglobin (HGB) level. In addition, for this study, we re-determined the pathological disease stage of each of the tumors in the KMC cohort based on the 8th edition of the international association for the study of lung cancer staging system [23]. Moreover, the KMC pathologist re-determined in each of the tumors the tumor differentiation grade and whether there was pleural or lympho-vascular invasion. 

### 2.4. Surgical Procedures and Follow-up

Of 61 cases, 48 patients had lobectomy and 13 patients had sub-lobar resection (12 wedge resections and 1 segmentectomy). Systematic lymph node dissection was performed in the majority of cases (50/61 of cases = 82%). Patients were seen at the clinics of the department of general thoracic surgery at KMC every 6 months during the first two postoperative years, and yearly thereafter. A low dose non-contrast CT scan of the chest was obtained prior to each clinic visit. Recurrence was defined as recurrent tumor in the lungs, pleural space, hilar or mediastinal lymph nodes and as metastasis outside the thorax. Overall survival (OS) was calculated from the day of surgery until the date of death or last follow-up. Disease-free survival (DFS) was measured from the date of surgery until an abnormal imaging test was detected during follow-up. The median OS and DSF times were 44 (interquartile range (IQR) 31 to 68) months and 39 (IQR 24 to 66) months, respectively.

### 2.5. CDC25C Immunostaining and Scoring of CDC25C Staining

Immunohistochemistry—antigen retrieval was performed in EDTA buffer for 20 min in microwave. Next, tissue sections were blocked for 30 min with CAS block (Thermo Fisher Scientific, Waltham, MA, USA) at room temperature, and then stained with the anti-human CDC25C monoclonal antibody (E302) (abcam, Cambridge, MA 02139-1517, USA) at a concentration of 2.1 µg/mL. Thereafter, the stained sections were incubated with HRP goat anti-rabbit (Agilent, Santa Clara, CA, USA) for 60 min. 3-amino-9-ethylcarbazole (AEC) was used for color development and sections were counterstained with hematoxylin.

Scoring of CDC25C staining—to quantify CDC25C staining, two independent coders, blind to clinical data, graded in each sample the intensity of CDC25C staining and the percentage of tumor cells staining positive for CDC25C. In particular, CDC25C staining intensity was graded from 0 to 3 (0—no staining, 1—weak intensity, 2—medium intensity and 3—high intensity) and the percentage of tumor cells showing positive CDC25C staining was graded from 0 to 100% in 12.5% intervals (i.e., 0%, 12.5%, 25%........ 87.5% and 100%). Next, we generated an H-score by multiplying the staining intensity by the percentage of tumor cells staining positive for CDC25C (H-score range 0 to 300). The H-score was then used to group the tumors into low, medium and high CDC25C-expressing tumors. The cut-off defining low, medium and high CDC25C staining was an H score of 0 to 37.5, 50 to 135 and 150 to 300, respectively.

### 2.6. Statistical Analysis

Categorical variables were expressed as absolute numbers and were analyzed using a Chi square test or a Fisher exact test as appropriate. The Kaplan–Meier method was used to compute actuarial OS and DFS. Differences in survival between groups were analyzed using the Log-rank test and the Gehan–Breslow–Wilcoxon test. Statistical analyses were performed using GraphPad Prism 9.0.2 software (GraphPad Software, 2365 Northside Dr. Suite 560, San Diego, CA 92108, USA).

## 3. Results

### 3.1. CDC25C Expression in the LC Tissue Array

To initially evaluate the expression of CDC25C in NSCLC, we stained the biomax LC807b LC tissue array for CDC25C. Our analysis focused on 27 LUAD and 31 LUSC tumor samples, whose pathological characteristics are presented in Table 1. We assigned low, medium and high CDC25C staining scores to 13, 6 and 8 of the LUAD tumors, respectively, and to 13, 9 and 9 of the LUSC tumors, respectively. We did not detect virtually any CDC25C staining in normal lung tissue. Representative staining of CDC25C in LUAD and LUSC tumors is shown in Figure 1A. Next, we searched for correlations between CDC25C expression levels and the tumor pathological characteristics. First, we analyzed the joint group of LUAD and LUSC tumors (*n* = 58). We found that high CDC25C expression was more common among poorly differentiated tumors, whereas low CDC25C expression was more common among well-differentiated tumors (*p* < 0.05) (Figure 1B). Second, we separately tested the correlation between the expression levels of CDC25C and the differentiation grade of the LUAD and LUSC tumors. We found that this correlation was nearly completely preserved in LUAD tumors (*p* = 0.056) but that it was completely lost in LUSC tumors (*p* > 0.6). Notably, we did not find any association between CDC25C expression and disease stage.

### 3.2. CDC25C Expression in the KMC LUAD Cohort

To further evaluate the expression of CDC25C in LUAD, we stained the KMC cohort tissue sections for CDC25C. The clinical characteristics of the patients in the cohort and the pathological characteristics of their tumors are presented in Table 2. We assigned low, medium and high CDC25C staining scores to 20, 20 and 21 of the tumors, respectively. Representative staining of CDC25C in well and poorly differentiated LUAD tumors is shown in Figure 2A. As demonstrated in Figure 2B, CDC25C expression in LUAD was mainly restricted to tumor cells, whereas areas of inflammation inside the tumor and tumor-adjacent lung parenchyma nearly did not stain for CDC25C. Notably, similar patterns of staining were also observed in the LC tissue array. When testing the correlations between CDC25C expression levels and the tumors pathological characteristics, we found that high CDC25C expression was more common among poorly differentiated tumors, whereas low CDC25C expression was more common among well-differentiated tumors (*p* < 0.0005) (Figure 2C). In addition, we found that high CDC25C expression was more common among tumors larger than 1cm (T1b and up), whereas low CDC25C expression was more common among tumors smaller than 1cm (T1a) (*p* < 0.05) (Figure 2D). No significant correlation between CDC25C expression levels and lymph node metastatic status, pleural invasion or lympho-vascular invasion was found. Finally, we compared the OS and the DFS of patients whose tumors expressed either high or low levels of CDC25C. As shown in Figure 3A, the OS of patients whose tumors expressed high CDC25C levels was inferior to the OS of patients whose tumors expressed low CDC25C levels, but this analysis was only marginally significant (*p* = 0.08). With respect to DFS, as shown in Figure 3B, patients whose tumors expressed high CDC25C levels had a significantly shorter DFS relative to patients whose tumors expressed low CDC25C levels (*p* < 0.05) (Figure 3B). Furthermore, the tumor recurrence rate among patients with pathological stage I disease (*n* = 30) was significantly lower in patients whose tumors expressed low levels of CDC25C (*p* < 0.05) (Figure 3C). No correlation between disease stage and CDC25C expression levels was found in the KMC cohort.

## 4. Discussion

In the current study, we test whether CDC25C protein detected and quantified by immunohistochemistry can serve as a prognostic marker in LUAD. We focused on CDC25C because past research has shown that CDC25C mRNA levels in LUAD correlate with clinical outcomes and with response to therapy [11,22]. Dovetailing with past work at the mRNA level, our findings demonstrate that CDC25C can indeed be detected in LUAD tumor sections by immunohistochemistry and that high CDC25C expression levels in the tumors are associated with aggressive histological features and with poor clinical outcomes.

Our study advances the development of CDC25C as a prognostic marker in LUAD in several aspects. First, our study is the first to test the link between LUAD tumor differentiation grade and the expression of CDC25C in the tumor. We demonstrate that well-differentiated LUAD tumors express low levels of CDC25C, whereas poorly differentiated LUAD tumors express high levels of CDC25C, thereby linking the expression of CDC25C with aggressive histological features of the tumor. Second, our study is also the first to test the link between CDC25C protein expression levels in early-stage LUAD and tumor size. In line with data extracted from the TCGA showing that mRNA levels of CDC25C were lower in T1 relative to T2 to T4 tumors, our study nicely demonstrates that, also, at the protein level, expression of CDC25C in LUAD was lower in tumors that were smaller than 1cm relative to larger tumors [22]. In contrast to data from the TCGA, we did not detect an association between CDC25C protein expression levels and disease stage; however, this is most probably due to the high representation (72%) of stage I disease in our cohort [22]. We contemplate that the concordance between the expression of CDC25C mRNA in LUAD and the expression of CDC25C protein in LUAD suggests that associations between CDC25C mRNA levels in LUAD and clinical outcomes could also be detected by quantifying CDC25C protein level in the tumor. In addition, indeed, in line with data from the TCGA, we report on a trend for better OS and on a significantly better DFS among LUAD patients that had low relative to high expression of CDC25C in their tumors. We further demonstrate that, among patients with stage I disease, recurrence rates were significantly lower in patients whose tumors expressed low levels of CDC25C. Finally, in contrast to previous research which has measured CDC25C expression in LUAD only using small core tissue samples, which do not necessarily represent the tumor that they were extracted from, we measured CDC25C staining in entire tumor tissue sections that were specifically selected for this study by the KMC pathologist [24]. As such, our study comprehensively documents CDC25C protein expression in distinct areas of the tumor. In particular, we show that CDC25C expression in LUAD is mainly restricted to tumor cells, whereas areas of inflammation inside the tumor and tumor-adjacent lung parenchyma nearly do not stain for CDC25C (Figure 2B). Together, these findings provide reasonable evidence in favor of developing CDC25C as a prognostic marker in LUAD. 

When searching the literature for previous works testing the correlation between CDC25C expression and cancer outcomes, we came across several studies; however, nearly all of them based their analysis on CDC25C mRNA levels. For example, CDC25C mRNA was shown to be elevated in hepatocellular carcinomas relative to nontumorigenic liver tissue and high CDC25C mRNA levels predicted poor outcomes [25]. Similarly, in pancreatic adenocarcinoma [26] and in breast cancer, high CDC25C mRNA levels indicated bad prognosis [27]. Interestingly, the sole study that we found which tested the correlation between CDC25C protein expression and cancer outcomes was performed in NSCLC by the group of Wang et al. (2015). Specifically, Wang and colleagues tested the association between the expression levels of CDC25C, among other cell cycle proteins, and the clinical outcomes of a joint cohort of stage III and IV, LUAD and LUSC patients that received at least two cycles of docetaxel and cisplatin doublet chemotherapy. In contrast to our findings, Wang et al. did not find a correlation between CDC25C protein expression levels and disease outcomes [24]. However, given that our investigation significantly differs from the study of Wang et al. in the study population (we tested CDC25C expression in a homogenous group of tumors that were resected from early-stage operable LUAD patients who did not receive any preoperative therapy), this dis-concordance is not surprising. It is evident that both studies highlight the need for further research in the field. 

Looking forward, given that, in the TCGA database, the linear association between CDC25C mRNA expression levels and the OS of early-stage (stage Ia to IIb) NSCLC patients is significant only among LUAD but not among LUSC patients (data not shown), it is evident that future research will have to separately measure the prognostic yield of CDC25C staining in distinct histological subtypes of NSCLC. Support for this assertion comes from our observations showing that, in the LC array, the association between higher CDC25C expression levels and poor NSCLC differentiation grade was attributed to LUAD rather than LUSC tumors.

Our study has several limitations. First, it is a retrospective study that is based on a relatively small series of patients that were operated over a relatively long period of time. Second, although all patients in the study were considered based on clinical staging to have upfront respectable early-stage LUAD, postoperative pathological staging indicated that, in 18% and 10% of patients, the disease stage was actually stage IIA/IIB and stage IIIA/IIIB, respectively, thus limiting the homogeneity of our cohort. Third, detailed preoperative clinical staging of the patients is missing. Nevertheless, our prospective data collection using the KMC departmental database and our complete follow-up rate on 100% of patients assures that the quality of clinical data used in this study is high. Furthermore, the comprehensive re-evaluation of each of the histological sections of the tumors in the cohort by the KMC pathologist assures both that accurate pathological staging was determined based on the eighth edition of the international association for the study of lung cancer staging system [23] and that the tumor histological grade was accurate. Finally, given that a single surgical team performed all surgeries, the unity in surgical skills and judgment is assured throughout the study. 

## 5. Conclusions

In the current study, we show that high expression of CDC25C protein in LUAD was predominant among poorly differentiated LUAD (*p* <0.001) and in LUAD > 1cm (*p* < 0.05). In addition, we report that high expression of CDC25C was associated with reduced disease-free survival (*p* = 0.03) and with a trend for reduced overall survival. Therefore, high expression of CDC25C in LUAD is associated with aggressive histological features and with poor outcomes. Given these observations and taking into consideration the prognostic value of CDC25C mRNA levels in LUAD, it is evident that larger studies are required to further validate CDC25C protein expression detected by immunohistochemistry as a prognostic marker in early-stage LUAD.

## Figures and Tables

**Figure 1 biomedicines-11-00362-f001:**
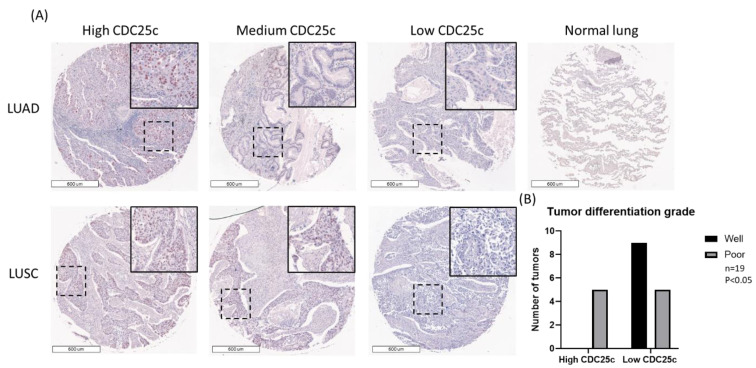
Patterns of CDC25C expression and correlation with tumor differentiation grade in the Biomax LC807b tissue array. (**A**) High, medium and low expression of CDC25C in LUAD and LUSC samples. Original magnification ×40. Normal lung tissue showing no CDC25C staining is also shown in A. (**B**) The number of well and poorly differentiated LUADs and LUSCs expressing either low or high levels of CDC25C is shown (*p* < 0.05). Abbreviations: CDC25C, cell division cycle 25C; LUAD, lung adenocarcinoma; LUSC, lung squamous cell carcinoma.

**Figure 2 biomedicines-11-00362-f002:**
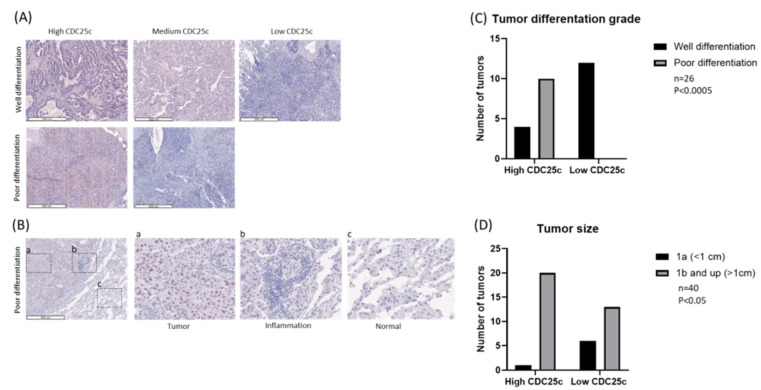
CDC25C expression and correlation with tumor differentiation grade and with tumor size in the KMC cohort. (**A**) High, medium and low expression of CDC25C in well and poorly differentiated LUAD samples. Original magnification ×40. (**B**) CDC25C expression in LUAD tumor parenchyma, in areas of inflammation inside the tumor and in LUAD-adjacent lung tissue is shown. (**C**) The number of well and poorly differentiated LUAD tumors expressing either low or high levels of CDC25C is shown (*p* < 0.0005). (**D**) The number of small (T1a—up to 1 cm) and large (T1b—1 cm and up) LUAD tumors expressing either low or high levels of CDC25C is shown (*p* < 0.05). Abbreviations: CDC25C, cell division cycle 25C; KMC, Kaplan medical center; LUAD, lung adenocarcinoma.

**Figure 3 biomedicines-11-00362-f003:**
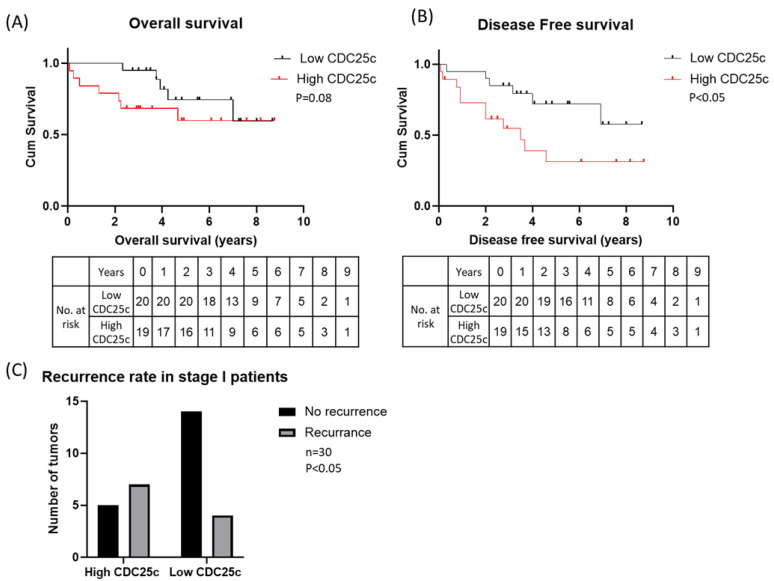
Long-term outcomes of LUAD patients according to CDC25C expression levels in the tumor. Kaplan–Meier curves for the OS (**A**) and DFS (**B**) of patients whose tumors express either low (black line) or high (red line) levels of CDC25C are shown. The number of patients at risk is shown below each graph. (**C**) The number of patients with stage I LUAD whose disease recurred or did not recur among patients with either low or high expression of CDC25C in their tumors is presented (*p* < 0.05). Abbreviations: OS = overall survival; DFS = disease-free survival; CDC25C = cell division cycle 25C.

**Table 1 biomedicines-11-00362-t001:** Clinical and pathological characteristics of 27 LUAD and 31 LUSC tumors included in the Biomax LC807b lung cancer tissue array. The available clinical and pathological characteristics of LUAD and LUSC tumors included in the Biomax LC807b lung cancer tissue array are presented.

Patient Characteristics	
	Cases (*n* = 58)
Age (years)	
Mean +/− SD	57.06 +/− 10.38
Sex	
Male/Female	46(79.3%)/12 (20.7%)
Pathological Characteristics	
Pathology	
LUAD/LUSC	27(46.5%)/31 (53.5%)
Pathological T	
1	5 (8.6%)
2	49 (84.5%)
3	4 (6.9%)
Pathological N	
0	35 (60.3%)
1	18 (31%)
2	5 (8.6%)
Pathological stage	
Stage I (IA/IB)	34 (58.6%)
Stage II (IIB)	18 (31%)
Stage III (IIIA)	6 (10.3%)
Tumor differentiation	
Well	14 (24.1%)
Moderate	28 (48.3%)
Poor	14 (24.1%)
N/A	2 (0.3%)

Abbreviations: LUAD = lung adenocarcinoma; LUSC = lung squamous cell carcinoma; SD = standard deviation.

**Table 2 biomedicines-11-00362-t002:** Clinical and pathological characteristics of patients and tumors in the KMC LUAD cohort. The clinical and pathological characteristics of patients in the KMC LUAD cohort are presented.

Patient Characteristics	
	Cases (*n* = 61)
Age (years)	
Mean +/− SD	70.07 +/− 8.23
Sex	
Male/Female	29 (47.5%)/32 (52.5%)
Smoking status	
Ever/Never	36 (59%)/25 (31%)
FEV1% of predicted	
Mean +/− SD	75% +/− 20%
Comorbidities	
COPD	21 (34.4%)
DM	16 (26.2%)
Hemoglobin level (gr%)	
Mean +/− SD	12.69 +/− 1.67
Pathological Characteristics	
Pathological T	
1a	16 (26.2%)
1b	14 (22.9%)
1c	7 (11.5%)
2a	15 (24.6%)
2b	2 (3.3%)
3	5 (8.2%)
4	2 (3.3%)
Pathological N	
0	51 (83.6%)
1	5 (8.2%)
2	5 (8.2%)
Pathological stage	
Stage I (IA1/IA2/IA3/IB)	44 (72.1%)
Stage II (IIA/IIB)	11 (18%)
Stage III (IIIA/IIIB)	6 (9.8%)
Tumor differentiation	
Well	24 (39.3%)
Moderate	22 (36%)
Poor	11 (18%)
N/A	4 (6.6%)
Pleural invasion	
Present/Absent	17 (27.9%)/44 (72.1%)
Lymphovascular invasion	
Present/Absent	10 (16.4%)/51 (83.6%)

Abbreviations: FEV1% = forced expiratory volume in 1 s % from predicted; COPD = chronic obstructive pulmonary disease; DM = diabetes mellitus; SD = standard deviation.

## Data Availability

The data presented in this study are available within this article.

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
