# Peer review of "CDC25C Protein Expression Correlates with Tumor Differentiation and Clinical Outcomes in Lung Adenocarcinoma"

_biomedicines, 2023, doi:10.3390/biomedicines11020362_

Round 1
Reviewer 1 Report
In this manuscript, the authors found that CDC25C protein in LUAD is associated with tumor differentiation and clinical outcomes. However, the following points should be satisfactorily addressed before the manuscript can be accepted for publication:
1, In the first part of the result, the authors didn’t distinguish the LU-Ads and LUSCs, is the trend of CDC25C in these two types of lung cancer the same, and why the authors focused on the LUSCs?
2, The authors did not find any association between CDC25C 183 expression and disease stage, Is this consistent with the published data?
Author Response
In this manuscript, the authors found that CDC25C protein in LUAD is associated with tumor differentiation and clinical outcomes. However, the following points should be satisfactorily addressed before the manuscript can be accepted for publication:
We thank the reviewer for thoroughly reading the manuscript and for raising important points that helped improve the manuscript.
1, In the first part of the result, the authors didn’t distinguish the LUADs and LUSCs, is the trend of CDC25C in these two types of lung cancer the same, and why the authors focused on the LUSCs?
This is an important issue. We address this issue in lines 184 to 191 in the revised version. Specifically, we report that: “First, we analyzed the joint group of LUAD and LUSC tumors (n=58). We found that high CDC25C expression was more common among poorly differentiated tumors whereas low CDC25C expression was more common among well differentiated tumors (p < 0.05) (Figure 1B). Second, we separately tested the correlation between the expression levels of CDC25C and the differentiation grade of the LUAD and LUSC tumors. We found that this correlation was nearly completely preserved in LUAD tumors (p = 0.056) but that it was completely lost in LUSC tumors (p > 0.6)”.
Base on this observation we in the second part of the results focus on LUAD.
2, The authors did not find any association between CDC25C expression and disease stage, Is this consistent with the published data?
We thank the reviewer for mentioning this key issue. We address this issue in the discussion section in lines 274 to 276 specifically explaining that “In contrast to data from the TCGA, we did not detect an association between CDC25C protein expression levels and disease stage however, this is most probably due to the high representation (72%) of stage I disease in our cohort”
Reviewer 2 Report
The present manuscript entitled “CDC25C protein expression correlates with tumor differentiation and clinical outcomes in lung adenocarcinoma” by Stern E. et al., describe the issues, a lung cancer (LC) tissue array and a cohort of 61 LUAD tissue sections were stained for CDC25C and the staining was blindly scored. High expression of CDC25C protein in LUAD is associated with aggressive histological features and with poor outcomes. The authors report an interesting approach. The objective and justification of the work are clear, and the experimental work is significant. The study is very accurate and adequate, and thus, I recommend it for publication. However, certain Minor issues are detailed below which need to be addressed before its final acceptance in Biomedicines.
Comment 1: There are so many typographical errors in the manuscript text, authors must check typos, use of is, are, was, were, and prepositions.
Comment 2: The abstract is poorly written, it should clearly discuss the problem statement and the current study approach. So it should be revised.
Comment 3: The introduction provided sufficient information with relevant details and references. However, I think the authors need to cite and discuss some more recent references in the introduction section to strengthen the section.
Comment 4: In line number 96, check to mention whether CDC25 or CDC25C
Comment 5: In line number 114 and 115 the preparation of the tissue section is mentioned, but the processing and staining details are missing. Provide the details.
Comment 6: In line number 118 the approval number of the study is mentioned, whereas the human ethical clearance committee permission details have not been mentioned. Specify the reason and include the details.
Comment 7: In line 186 delete the hyphen in the word array
Comment 8: The results need to discuss from multiple angles and discussed in the context of previous literature
Comment 9: The major criticism of the manuscript is that it has not discussed the connection of CDC25C with cancer. Only it has been discussed in line 284 that the present study differs from Wang et al., 2015 study.
Comment 10: The conclusions section is too short, the authors should revise it.
Author Response
The present manuscript entitled “CDC25C protein expression correlates with tumor differentiation and clinical outcomes in lung adenocarcinoma” by Stern E. et al., describe the issues, a lung cancer (LC) tissue array and a cohort of 61 LUAD tissue sections were stained for CDC25C and the staining was blindly scored. High expression of CDC25C protein in LUAD is associated with aggressive histological features and with poor outcomes. The authors report an interesting approach. The objective and justification of the work are clear, and the experimental work is significant. The study is very accurate and adequate, and thus, I recommend it for publication. However, certain Minor issues are detailed below which need to be addressed before its final acceptance in Biomedicines.
We thank the reviewer for thoroughly reading the manuscript and for finding merit in our work. We are further grateful for the important comments that the reviewer raised which helped to significantly improve the manuscript.
Comment 1: There are so many typographical errors in the manuscript text, authors must check typos, use of is, are, was, were, and prepositions.
We thank the reviewer for thoroughly reading the manuscript. We have reviewed the manuscript carefully aiming to address all typographical errors in the text.
Comment 2: The abstract is poorly written, it should clearly discuss the problem statement and the current study approach. So it should be revised.
In line with the reviewer’s suggestion the abstract of the manuscript was completely revised to streamline its reading and comprehensibility.
“Given that even after multimodal therapy, early-stage lung-cancer (LC) often recurs, novel prognostic markers to help guide therapy are highly desired. The mRNA levels of Cell Division Cycle 25C (CDC25C), a phosphatase that regulates G2/M cell cycle transition in malignant cells, corelate with poor clinical outcomes in lung adenocarcinoma (LUAD). However, whether CDC25C protein detected by immunohistochemistry can severe as a prognostic marker in LUAD is yet unknown. We stained a LC tissue array and a cohort of 61 LUAD tissue sections for CDC25C and searched for correlations between CDC25C staining score and the pathological characteristics of the tumors and the patient’s clinical outcomes. Clinical data was retrieved from our prospectively maintained departmental database. We found that high expression of CDC25C was predominant among poorly differentiated LUAD (p <0.001) and in LUAD > 1cm (p < 0.05). Further, high expression of CDC25C was associated with reduced disease-free survival (p = 0.03, median follow up of 39 months) and with a trend for reduced overall survival (p = 0.08). Therefore, high expression of CDC25C protein in LUAD is associated with aggressive histological features and with poor outcomes. Larger studies are required to further validate CDC25C as a prognostic marker in LUAD.”
Comment 3: The introduction provided sufficient information with relevant details and references. However, I think the authors need to cite and discuss some more recent references in the introduction section to strengthen the section.
We thank the reviewer for this comment. To strengthen the introduction and to all in all expand the coverage of recent publications in the manuscript four new references were added to it. New reference number 21 as well as new reference numbers 25 to 27, all published between 2020 to 2022. Given the limited available literature on CDC25C and NSCLC in general and on CDC25C and LUAD in particular it was hard to significantly expand the introduction while keeping it focused. Nonetheless, the findings about the relation between CDC25C and the immune infiltration of LUAD tumors as described in reference 21 have now been added to the introduction - lines 82 to 87. Further the mentioning of the findings of reference 22 were also expanded - lines 87 to 90. Consequently, the introduction now more thoroughly describes the association between CDC25C mRNA and the immune milieu of LUAD tumors.
“More recently, two studies reported that CDC25C mRNA levels in LUAD influence the tumor immune cell infiltration as well as its response to immunotherapy. In particular, CDC25C mRNA levels were shown to positively corelate with the infiltration of Th2 T cells to the tumors while they were shown to negatively corelate with the infiltration of mast cells, eosinophils, natural killer cells and dendritic cells to the tumors [21]. In addition, high CDC25C mRNA levels in the tumor were shown to corelate with expression of genes that drive immune suppression. In line with these observations, elevated levels of CDC25C mRNA in LUAD were also linked to shorter progression-free survival in LUAD patients treated with immunotherapy [22]. From a clinical perspective, these reports suggest that CDC25C mRNA levels in LUAD may potentially guide therapeutic decisions.”
Comment 4: In line number 96, check to mention whether CDC25 or CDC25C
CDC25 was changed to CDC25C.
Comment 5: In line number 114 and 115 the preparation of the tissue section is mentioned, but the processing and staining details are missing. Provide the details.
Tissue collection is indeed described in the Materials and Methods section 2.2 in lines 118 to lines 121.
“The pathologist specifically selected slides that contained both tumor tissue and tumor adjacent lung parenchyma. We prepared tissue section from the formalin fixed and paraffin embedded tumor blocks that corresponded to the pathologist selection and we then stained them for CDC25C”
The exact tissue processing and CDC25C staining procedure is thoroughly described in the Materials and Methods section 2.5 lines 150 to 156.
“Immunohistochemistry – antigen retrieval was performed in EDTA buffer for 20 minutes in microwave. Next, tissue sections were blocked for 30 minutes with CAS block (life technologies) at room temperature and then stained with the anti-human CDC25C monoclonal antibody [E302] (abcam, Cambridge, MA 02139-1517, USA) at a concentration of 2.1 µg/ml. Thereafter, the stained sections were incubated with HRP goat anti Rabbit (dako), for 60 minutes. 3-amino-9-ethylcarbazole (AEC) was used for color development, and sections were counterstained with hematoxylin.”
Comment 6: In line number 118 the approval number of the study is mentioned, whereas the human ethical clearance committee permission details have not been mentioned. Specify the reason and include the details.
The precise details of the human ethical clearance committee permission have been clarified in the text – lines 123 to 126.
“The Kaplan Hebrew University Medical Center, institutional review board (institutional Helsinki committee) approved this study (protocol number KMC-0233-17). The review board permitted acquisition of surplus histological material for analysis and review of patient’s medical records.”
Comment 7: In line 186 delete the hyphen in the word array
The hyphen in the word array was deleted
Comment 8: The results need to discuss from multiple angles and discussed in the context of previous literature
This is an important issue; we thank the reviewer for raising it. Unfortunately, we did not find aside of the study of Wang et al. other studies that tested the association between CDC25C protein expression and cancer outcomes in general an NSCLC in particular. We have however found additional studies that tested the association between CDC25C mRNA expression and cancer outcomes. In line with the reviewer comments, we have expanded on these studies in the discussion. By doing so, we aimed to provide a more holistic view on the potential of CDC25C to serve as a cancer prognostic marked. Please see more details in the response to comment 9.
Comment 9: The major criticism of the manuscript is that it has not discussed the connection of CDC25C with cancer. Only it has been discussed in line 284 that the present study differs from Wang et al., 2015 study.
This is an important issue; we thank the reviewer for raising it. The role of CDC25C as a prognostic marker in cancer in general and in NSCLC in particular is more thoroughly discussed in the revised discussion. The need to verify the prognostic yield of CDC25C not only at the mRNA level but also at the protein level is clearer in the revised section while this section also highlights the brother implications of CDC25C as a prognostic marker in other cancers. The revised discussion also highlights the importance of testing the prognostic yield of CDC25C in distinct subtypes of NSCLC. Lines 294 to 319.
“When searching the literature for previous works testing the correlation between CDC25C expression and cancer outcomes we came across several studies however, nearly all of them based their analysis on CDC25C mRNA levels. For example, CDC25C mRNA was shown to be elevated in hepatocellular carcinomas relative to non-tumorigenic liver tissue and high CDC25C mRNA levels predicted poor outcomes [25]. Similarly, in pancreatic adenocarcinoma [26] and in breast cancer, high CDC25C mRNA levels indicated bad prognosis [27]. Interestingly, the sole study that we found which tested the correlation between CDC25C protein expression and cancer outcomes was performed in NSCLC by the group of Wang et al. (2015). Specifically, Wang an colleagues tested the association between the expression levels of CDC25C, among other cell cycles proteins, and the clinical outcomes of a joint cohort of stage III and IV, LUAD and LUSC patients that received at least two cycles of docetaxel and cisplatin doublet chemotherapy. In contrast to our findings, Wang et al. did not find a correlation between CDC25C protein expression levels and disease outcomes [24]. However, given that our investigation significantly differs from the study of Wang et al. in the study population (we tested CDC25C expression in a homogenous group of tumors that were resected from early stage operable LUAD patients who did not receive any preoperative therapy), this dis-concordance is not surprising. It is evident that both studies highlight the need for further research in the field.
Looking forward, given that in the TCGA database, the linear association between CDC25C mRNA expression levels and the OS of early stage (stage Ia to IIb) NSCLC patients is significant only among LUAD but not among LUSC patients (data not shown), it is evident that future research will have to separately measure the prognostic yield of CDC25C staining in distinct histological subtypes of NSCLC. Support for this assertion comes from our observations showing that in the LC array, the association between higher CDC25C expression levels and poor NSCLC differentiation grade was attributed to LUAD rather than LUSC tumors.”
Comment 10: The conclusions section is too short, the authors should revise it.
The conclusion section has been expanded – lines 336 to 344.
“In the current study, we show that high expression of CDC25C protein in LUAD was predominant among poorly differentiated LUAD (p <0.001) and in LUAD > 1cm (p < 0.05). In addition, we report that, high expression of CDC25C was associated with reduced disease-free survival (p = 0.03) and with a trend for reduced overall survival. Therefore, high expression of CDC25C in LUAD is associated with aggressive histological features and with poor outcomes. Given these observations, and taking into consideration the prognostic value of CDC25C mRNA levels in LUAD, it is evident that larger studies are required to further validate CDC25C protein expression detected by immunohistochemistry as a prognostic marker in early stage LUAD.”
Reviewer 3 Report
I have gone through the manuscript. Topic is indeed interesting and meets the requirements for consideration as a diagnostic and prognostic marker. Authors should address preclinical aspects as well. CDC25C should be tested in cell culture and animal model studies to validate the mechanistic involvement of CDC25C in cancer regulation. Another important aspect could be to test the patient derived xenografts against different chemotherapeutic drugs. Overall different clues of information will make the claims of these researchers more strong and credible.
Author Response
I have gone through the manuscript. Topic is indeed interesting and meets the requirements for consideration as a diagnostic and prognostic marker. Authors should address preclinical aspects as well. CDC25C should be tested in cell culture and animal model studies to validate the mechanistic involvement of CDC25C in cancer regulation. Another important aspect could be to test the patient derived xenografts against different chemotherapeutic drugs. Overall different clues of information will make the claims of these researchers more strong and credible.
We thank the reviewer for thoroughly reading the manuscript and for finding merit in our work. We absolutely agree with the reviewer that preclinical aspects considering the role of CDC25C in lung cancer pathogenesis is an important topic of research. The proposal to use cell culture, animal models and patient derived xenografts to mechanistically test the involvement of CDC25C in cancer regulation is certainly valid. Nonetheless, the key focus of our investigation was to test whether CDC25C, detected by immunohistochemistry, may serve as a prognostic marker in early-stage NSCLC. Our investigation shows that in LUAD rather than in LUSC CDC25C may potentially be a novel prognostic marker. In depth investigation of the mechanism through which CDC25C may promote NSCLC propagation, though super interesting and important, is beyond the scope of our current study which was designed to be more clinically rather than basic science oriented.
We hope that within the limits our this study the reviewer will consider the merit of our findings sufficiently important to justify its publication. We once again thank the reviewer for the time, effort and thought that he put in reading and commenting on our manuscript.
Round 2
Reviewer 3 Report
Looks better now